# SeMa3D: Lifting Vision-Language Models for Unsupervised 3D Semantic Correspondence

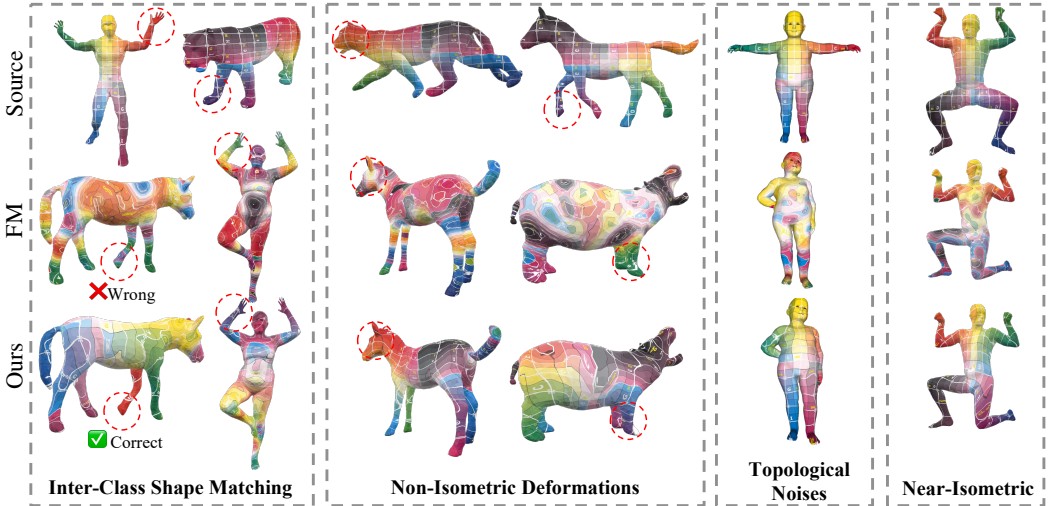

Figure 1: SeMa3D **consistently achieves high-quality dense correspondences** rather than functional map baseline (Cao et al., 2023) under different challenging settings: 1) *Inter-class shape matching*: our method "understands" semantic meanings of input shapes, rather than solely geometric properties, achieving cross-category semantic consistency. Each part, like "hand" and "leg" of the human, is aligned with "front leg" and "back leg" of the horse, respectively. 2) *Non-isometric deformations*: our method achieves cross-instance semantic consistency for non-isometric deformations. 3) *Topological noises and near-isometric deformations*: our method produces high-quality mapping against topological noises and near-isometric deformations, respectively.

## ABSTRACT

We tackle unsupervised dense semantic correspondence for 3D shapes, focusing on severe **non-isometric** deformations and **inter-class** matching–a regime where conventional functional map pipelines fail due to ambiguous geometric cues. We propose *SeMa3D*, a framework that integrates semantic knowledge from vision-language foundation models to build robust vertex-level descriptors. Specifically, SeMa3D aggregates multi-view features from visual foundation models, with a novel *colorization* strategy that mitigates semantic inconsistencies across renderings, and further enriches them with *text embedding fields* to capture higher-level information. These descriptors are fused with geometric priors and aligned through a functional map formulation to ensure smooth, globally consistent correspondences. To achieve semantic matching, we introduce a *region-aware contrastive loss* that leverages geodesic distances and zero-shot semantic part proposals (*e.g.*, head, leg), injecting structural intent (*e.g.*, "head→head") into the mapping. Extensive experiments on challenging benchmarks show that SeMa3D outperforms existing methods in both extreme non-isometric and inter-class scenarios, achieving strong accuracy and generalization without relying on 3D labels or category-specific training.

# 1 INTRODUCTION

Finding dense correspondences between 3D shapes is critical for various vision tasks, such as shape analysis (Bogo et al., 2014), texture mapping (Ezuz & Ben-Chen, 2017), robotic manipulation (Zhu et al., 2025), and interpolation (Eisenberger et al., 2021). Early approaches are mainly optimization-based, which solve the problem by minimizing a predefined energy function and thus often encounter complex optimization problems (Van Kaick et al., 2011). Recent learning-based approaches Lang et al. (2021); Zeng et al. (2021) learn robust descriptors and correspondences using deep architectures such as DGCNN (Wang et al., 2019) and DiffusionNet (Sharp et al., 2022). Among the deep learning approaches, a line of works leverage functional map to establish correspondences by representing mappings between function spaces on shapes as compact matrices in the spectral domain, demonstrating competitive accuracy without manual annotations on near-isometric shape matching (Litany et al., 2017; Roufosse et al., 2019; Donati et al., 2020; Cao et al., 2023). Despite the progress, they often focus on near-isometric deformations and intra-class shapes and struggle to distinguish geometrically dissimilar but semantically similar parts due to the absence of semantic information. To this end, calculating dense correspondences between extremely non-isometric and inter-class shapes continues to be highly challenging, while less attention has been paid to them.

Recently, vision foundation models (VFMs) (Radford et al., 2021; Zhai et al., 2023) and vision-language models (VLMs) (Caron et al., 2021; Oquab et al., 2023; Rombach et al., 2022) have shown remarkable generalization as feature extractors for tasks like image matching (Zhang et al., 2023a), semantic segmentation (Li et al., 2023), and visual grounding (Liu et al., 2024a). Inspired by this, contemporary works (Dutt et al., 2024; Abdelreheem et al., 2023a; Zhu et al., 2025) employ 2D visual features from VFMs to improve non-isometric shape matching, establishing semantic correspondences (Dutt et al., 2024; Zhu et al., 2025; Abdelreheem et al., 2023a). While effective, these methods remain limited: (i) *multi-view features are extracted independently, leading to cross-view inconsistency (You et al., 2024)*; (ii) *semantics are restricted to the visual domain, neglecting language cues shown useful for 3D reasoning (Qin et al., 2024; Stojanov et al., 2025)*; and (iii) *the absence of explicit semantic relations between shape pairs hinders robust dense correspondences, especially across classes.*

To bridge these gaps, we propose a novel framework, **SeMa3D**, to exploit existing VLMs for the extraction of semantic features from visual and linguistic domains. First, we perform texture synthesis for raw 3D shapes (Liu et al., 2024b) and zero-shot semantic segmentation on a predefined part proposal to obtain semantic regions (Abdelreheem et al., 2023b). Second, we adopt VLMs to extract view-consistent semantic features from multiple renderings, including both visual and linguistic domains, and project them from 2D images onto 3D meshes to obtain per-vertex features. Finally, we distill the strengths of VLMs into the functional map framework, attached with visual features and language embeddings, ensuring smooth and high-quality mappings. Despite the constraints of the functional map framework, we explicitly enforce semantic region consistency between the input shape pair given the part proposal via a carefully-chosen contrastive loss, *i.e.*, encouraging vertex features from the same part to be similar to vertex features from different parts.

Empowered by novel components, SeMa3D empirically outperforms both semantic feature-based and functional map-based approaches in all scenarios, including near-isometric, non-isometric, and inter-class shape matching. In summary, we make the following contributions: 1) We present SeMa3D, extracting semantic features from VFMs that eliminates the ineffectiveness in non-isometric shape matching rather than conventional functional map pipeline; 2) We propose a *colorization* strategy that mitigates inconsistencies in semantic features extracted from multi-view renderings; 3) We enrich visual features with *text embedding fields* from VLMs, injecting higher-level semantic cues into the descriptors; 4) We design a region-aware contrastive loss to align semantic relations across shapes, enabling robust inter-class and non-isometric correspondences.

# 2 RELATED WORK

**Deep learning-based shape correspondences.** 3D shape matching aims to establish point-to-point correspondences between 3D shapes. Among existing methods, the functional map framework (Ovsjanikov et al., 2012) remains the dominant approach, leveraging spectral features and structural constraints. This framework has inspired numerous variants (Litany et al., 2017; Roufosse

et al., 2019; Donati et al., 2020; Cao & Bernard, 2023; Cao et al., 2023), with extensions focusing on functional map refinement (Melzi et al., 2019b) and cycle consistency (Cao & Bernard, 2022; Sun et al., 2023). However, most approaches rely on handcrafted geometric descriptors, such as the Wave Kernel Signature (WKS) (Aubry et al., 2011) and SHOT (Salti et al., 2014), or deep features extracted from input shapes (Cao & Bernard, 2023; Cao et al., 2023). Recent studies (Wimmer et al., 2024) suggest that traditional geometric features struggle with extreme cases and are sensitive to specific deformations, while incorporating semantic information can yield more robust and meaningful representations.

**VFMs for Semantic Correspondences.** The success of 2D VFMs like CLIP (Radford et al., 2021), DINO (Oquab et al., 2023), and diffusion models (Rombach et al., 2022) in image matching has inspired a new wave of 3D correspondence methods. These approaches lift powerful 2D semantic features onto 3D shapes via multi-view rendering to tackle challenging matching problems. Several recent works have pioneered this direction. *ZSC* (Abdelreheem et al., 2023a) uses VLMs and LLMs to generate coarse, zero-shot region proposals, which then initialize a functional map pipeline to find dense correspondences. *Diff3F* (Dutt et al., 2024) leverages pre-trained diffusion models to extract robust semantic descriptors from untextured shapes without any fine-tuning. Similarly, *Dense-Matcher* (Zhu et al., 2025) combines features from 2D foundation models with 3D network priors to compute dense matches, enabling tasks like cross-category color transfer. However, these methods are not effective for tackling inter-class and non-isometric shape matching. *ZSC* only uses semantic regions to initialize an axiomatic method (Ren et al., 2018) to get dense correspondences and ignore rich semantic features from VFMs. *Diff3F* utilizes diffusion models to "paint" untextured shapes, thereby extracting visual features that can cause inconsistency across multiple views. *DenseMatcher* combines the semantic features from VFMs and the functional map framework but relies on colored shape benchmarks and region annotations. To this end, our proposed method alleviates all the drawbacks of them through view-consistent colorization, zero-shot part segmentation, visual-linguistic semantic features from VLMs, and a dedicated contrastive loss to learn semantic priors.

## 3 METHOD

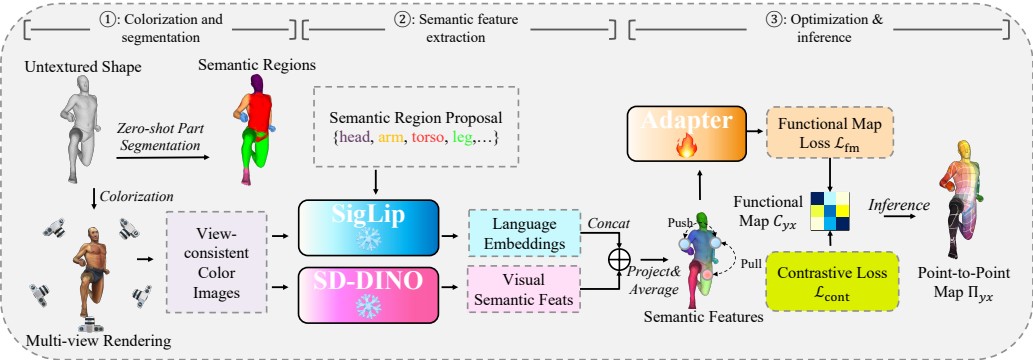

Figure 2: The overall structure of SeMa3D. Firstly, we perform high-quality texture synthesis and multi-view rendering. Next, we extract visual features (from SD-DINO (Zhang et al., 2023a)) and linguistic features (from SigLip (Tschannen et al., 2025)) and zero-shot part segmentation (Abdel-reheem et al., 2023b) to form the final semantic features. Finally, we adopt the functional map framework (Ovsjanikov et al., 2012) with a dedicated region-aware contrastive loss to compute dense correspondences.

### 3.1 PRELIMINARIES: NON-ISOMETRIC AND INTER-CLASS MATCHING

Most functional-map-based approaches target near-isometric deformations and *category-specific (intra-class)* matching (*e.g.*, human-to-human) (Cao & Bernard, 2022; 2023; Cao et al., 2023). In contrast, *inter-class* matching (*e.g.*, dog-to-human) remains underexplored (Dutt et al., 2024; Abdelreheem et al., 2023a) due to the isometry bias of functional maps (Ovsjanikov et al., 2012) and the limitations of purely geometric descriptors under large deformations (Wimmer et al., 2024).

To address the challenges of non-isometric deformations and inter-class shape matching, our key idea is to inject *semantic priors* into the correspondence process. We first coarsely align homologous *semantic regions* (*e.g.*, head, torso, limbs) across shapes, and then guide dense vertex-level correspondences with such semantic priors. To this end, we obtain region proposals, derive language embeddings for region names, and fuse them with VFM-derived visual features to construct semantic-aware vertex descriptors.

## 3.2 PIPELINE OVERVIEW

Following prior works (Cao et al., 2024; Dutt et al., 2024), given two shapes $\mathcal{X}$ and $\mathcal{Y}$ with $n_x$ and $n_y$ vertices, our objective is to learn a pointwise map $\Pi_{xy} \in \{0,1\}^{n_x \times n_y}$, where $\Pi_{xy}[i,j] = 1$ if and only if vertex $\mathcal{X}_i$ corresponds to vertex $\mathcal{Y}_j$. As shown in Fig. 2, our SeMa3D proceeds in four stages: (i) texture synthesis and multi-view rendering; (ii) view-consistent semantic feature extraction from VFMs and 3D back-projection; (iii) zero-shot semantic region proposal and language-augmented descriptors; (iv) optimization with a functional-map objective coupled with a semantic and region-aware contrastive loss.

## 3.3 VIEW-CONSISTENT SEMANTIC FEATURE EXTRACTION FROM VLMS

**Colorization for view-consistent visual features.** Since modern VFMs are trained on realistic color images, the absence of textures in most shape matching benchmarks poses a major challenge for semantic feature extraction. To this end, we adopt an off-the-shelf texture synthesis algorithm, SyncMVD (Liu et al., 2024b), to generate realistic textures for raw 3D shapes. We select SyncMVD over alternatives such as TEXTure (Richardson et al., 2023) because of its superior cross-view consistency, which is crucial in our setting. Compared with post-synthesis through diffusion models in Diff3F, SyncMVD produces realistic and view-consistent textures, enabling view-consistent and high-quality features, whereas Diff3F often yields noisy and artifact-prone results (see Fig. 3).

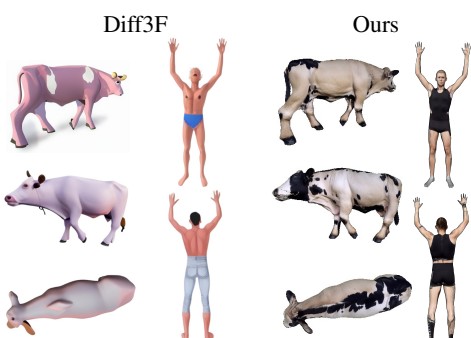

Figure 3: Comparison of view consistency between Diff3F and our colorization.

**Multi-view rendering.** We render $k$ views of each textured shape with uniformly distributed elevation and azimuthal angles in $[0°, 360°)$. Each shape is centered around the origin point and normalized to be inside a unit sphere. Given an input 3D mesh $\mathcal{X}$, the image projector $\mathcal{P}$ Dutt et al. (2024) with camera $\mathcal{C}_i$ is defined as

$$\mathcal{P}(\cdot|\mathcal{C}_i) := \mathcal{X} \mapsto I_i, \tag{1}$$

where $I_i$ denotes the image of height $H$ and width $W$ rendered from $\mathcal{X}$ by the $i$-th camera $\mathcal{C}_i$.

**Semantic feature extraction with VFMs.** We leverage modern, powerful VFMs for semantic feature extraction. Specifically, we extract visual semantic features using SD-DINO (Zhang et al., 2023a), with DINOv2 (Oquab et al., 2023) and Stable Diffusion (Zhang et al., 2023b; Rombach et al., 2022) as backbones. Since features extracted from VFMs are low-resolution and lack details, both features are enhanced to match the original image size using the FeatUp upscaler (Fu et al., 2024). We then fuse the two features to produce stronger semantic pixel-wise descriptors:

$$\mathcal{F}^{2D} = \alpha \mathcal{F}^{Diff} + (1-\alpha)\mathcal{F}^{DINO}, \tag{2}$$

where $\alpha$ is a parameter to control each component. As illustrated in Fig. 4, our method produces high-quality, distinguishable feature fields compared with Diff3F.

**Feature aggregation and 3D back-projection.** Given known camera parameters, we back-project 2D features into 3D space and aggregate them to obtain per-vertex descriptors. For each vertex $v_i$, we retrieve its semantic features from all visible views, and set it to zero if it is not observed. The back-projected features from $k$ views are then averaged at each vertex:

$$\mathcal{F}^{Sem-Vis} = \frac{1}{k}\sum_{i=1}^{k} \mathcal{P}^{-1}(\mathcal{F}_i^{2D}), \tag{3}$$

where $\mathcal{P}^{-1}$ is the back-projection operator with known camera parameters, *i.e.*, the inverse of Eq. 1.

## 3.4 SEMANTIC REGION PROPOSAL AND LANGUAGE EMBEDDING

As discussed in Sec. 3.1, we address non-isometric deformations and inter-class matching by introducing semantic understanding via *semantic regions*. We now describe how these regions are obtained and how language embeddings further enhance semantic features.

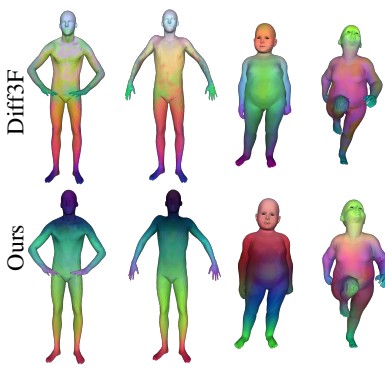

Figure 4: Feature field comparison between Diff3F and our SeMa3D.

**Semantic region proposal via zero-shot segmentation.** Unlike DenseMatcher (Zhu et al., 2025), which requires manual part annotations, we adopt a zero-shot approach, SATR (Abdelreheem et al., 2023b), for 3D part segmentation. It first processes multi-view renderings by GLIP (Li et al., 2022b) to generate bounding boxes according to a predefined semantic proposal. The proposal may be user-specified or derived from large language models (Abdelreheem et al., 2023a); in our setting, we use category-dependent part names such as torso, head, leg, and arm. Then SATR (Abdelreheem et al., 2023b) integrates *Gaussian geodesic reweighting* and *visibility smoothing*, to aggregate bounding boxes into final segmentations. This yields $n_{\mathcal{R}}$ disjoint semantic regions, with each vertex $v_i$ assigned a label $\mathcal{R}_i$. Unmatched vertices are assigned to the nearest region. Fig. 5 shows a segmentation example.

**Improve semantic features with language embeddings.** Unlike previous works that focus on visual-only semantic features Dutt et al. (2024); Zhu et al. (2025), we compute semantic region conditioned features with language cues to enable semantic region-awareness. To achieve this, we employ SigLip (Zhai et al., 2023; Tschannen et al., 2025) to fetch language embeddings for each semantic region by feeding part names. Specifically, for a vertex that belongs to the semantic region $\mathcal{R}_i$, we get the corresponding text embedding by:

$$\mathcal{F}_i^{\text{Sem-Lang}} = \texttt{SigLip}(\mathcal{R}_i). \tag{4}$$

In this way, vertices in the same semantic region share the same language embedding, while distinct regions have different linguistic features. Finally, we compute the final semantic features $\mathcal{F}^{\text{Sem}}$ by concatenating the visual feature $\mathcal{F}^{\text{Sem-Vis}}$ and the linguistic feature $\mathcal{F}^{\text{Sem-Lang}}$ for each vertex.

## 3.5 LEARNING SEMANTICS VIA CONTRASTIVE LOSS

Our optimization combines two objectives: a functional map loss to ensure smooth correspondences, and a semantic contrastive loss to enhance feature learning and embed region priors.

### 3.5.1 FUNCTIONAL MAP LOSS

Rather than computing point-wise correspondences based on feature similarities (Dutt et al., 2024; Zhang et al., 2023a), we adopt the widely used functional map framework to produce smooth and high-quality mappings (Cao et al., 2023; Ovsjanikov et al., 2012). Following the conventional deep functional map paradigm (Cao et al., 2023; Roufosse et al., 2019), we feed the semantic features $\mathcal{F}^{\text{Sem}}$ obtained from Section 3.4 into the DiffusionNet (Sharp et al., 2022) "adapter" to produce refined descriptors for further optimization:

$$\mathcal{F}_{\text{out}} = f_\theta(\mathcal{F}), \tag{5}$$

where $f_\theta$ is the trainable adapter. It is worth noting that $f_\theta$ is the only trainable part in our proposed framework. The refined features $\mathcal{F}_{\text{out}}$ are then used for calculating functional map losses. The optimal functional mapping $C_{yx}$ is calculated by performing two losses: data preserving loss $L_{\text{data}}$ to preserve input descriptors $\mathcal{F}_{\text{out}}$ and regularization loss $\mathcal{L}_{\text{reg}}$ to ensure mathematical properties like bijectivity and orthogonality. We also use the coupling loss $L_{\text{couple}}$ to ensure the consistency between soft correspondences (calculated by the cosine similarity of $\mathcal{F}_{\text{out}}$) and the functional map $C_{yx}$. The final functional map loss is obtained by:

$$\mathcal{L}_{\text{fm}} = \mathcal{L}_{\text{data}} + \lambda_{\text{reg}} \cdot \mathcal{L}_{\text{reg}} + \lambda_{\text{couple}} \cdot \mathcal{L}_{\text{couple}}. \tag{6}$$

Details of the functional map pipeline are described in Appendix A.

### 3.5.2 Semantic Region-Aware Contrastive Loss

**Semantic region and semantic structure.** To transfer prior relations between semantic regions, we need to measure the distance between them. Rather than using geodesic distance only, we propose to use the combination of semantic topology and geometric distances to measure the semantic distance between two regions. In detail, we first construct a semantic region graph to embed the region connectivity (Fig. 5). For two connected regions, we form a bipartite graph where the vertices represent the two regions and the edge weights are given by the average geodesic distance of an optimal graph match. Then, the optimal match $\pi$ is solved by the Jonker-Volgenant algorithm (Crouse, 2016). Formally, the semantic distance of two connected regions is calculated by:

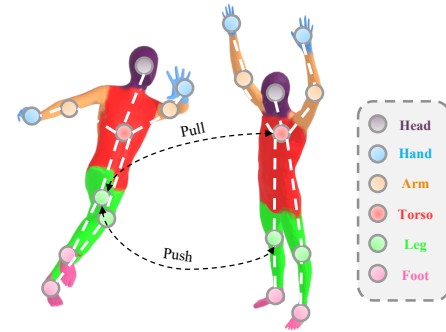

Figure 5: Example of semantic proposal and the corresponding region mapping.

$$\mathcal{D}_{\text{sem}}(\mathcal{R}_i, \mathcal{R}_j) = \begin{cases} 0, & \text{if } \mathcal{R}_i = \mathcal{R}_j \\ \frac{1}{n} \sum_{k=1}^{n} \mathcal{D}_{\text{geo}}(\pi(k)), & \text{if } \mathcal{R}_i \neq \mathcal{R}_j \end{cases},$$

where $n = \min(n_{\mathcal{R}_i}, n_{\mathcal{R}_j})$ is the number of matched pairs, and $\mathcal{D}_{\text{geo}}(\pi(k))$ denotes the geodesic distance of the $k$-th match. For unconnected regions (*e.g.*, head and leg), the semantic distance is defined as the shortest path length in the semantic region graph, which can be efficiently computed via Dijkstra's algorithm.

**Semantic region contrastive loss.** Given semantic regions, an intuitive solution to embed semantic information is to use InfoNCE loss (Xie et al., 2020). However, InfoNCE loss can not convey geometric closeness between regions, which is a characteristic of 3D data. In contrast, we introduce a modified contrastive loss to transfer prior relations between semantic regions. In detail, for a pivot vertex $v_i$ and another vertex $v_j$, we attract them if they belong to the same semantic region and repel them otherwise, according to the semantic distance between $\mathcal{R}_i$ and $\mathcal{R}_j$ with a dynamic margin $m_{\text{up}}$:

$$\mathcal{L}_{\text{cont}} = \begin{cases} \|\mathcal{F}_i - \mathcal{F}_j\|, & \text{if } \mathcal{D}_{\text{sem}}(\mathcal{R}_i, \mathcal{R}_j) = 0 \\ \texttt{ReLU}(m_{\text{up}} - \|\mathcal{F}_i - \mathcal{F}_j\|), & \text{if } \mathcal{D}_{\text{sem}}(\mathcal{R}_i, \mathcal{R}_j) \neq 0 \end{cases}, \tag{7}$$

where $m_{\text{up}}$ is an adaptive margin that controls the maximum dissimilarity between features of different regions. $m_{\text{up}}$ is defined proportionally to the semantic distance: $m_{\text{up}} = m_{\text{base}} \mathcal{D}_{\text{sem}}(\mathcal{R}_i, \mathcal{R}_j)$, with $m_{\text{base}}$ as a scaling parameter. This formulation offers three advantages. First, it enforces features from the same semantic region to be similar, which imposes the semantic region prior as additional supervision. Second, the distances of features from different semantic regions are maximized, up to an upper bound. Different from PointInfoNCE loss (Oord et al., 2018), which tries to pull negative pairs as far apart as possible, we stop optimizing the dissimilarities when they are far enough, since the segmentation results can be noisy. Third, the dynamic lower bound margin makes our loss flexible for diverse semantic structures of different benchmarks.

## 4 Experiments

In this section, we evaluate the performance of our SeMa3D and competing approaches under various challenging scenarios. We first evaluate performance on challenging inter-class shape matching in Section 4.1. Next, we perform experiments on non-isometric and near-isometric benchmarks in Section 4.2 and Section 4.3, respectively. Finally, we conduct ablation studies in Section 4.4.

**Evaluation protocol.** Following previous works (Cao & Bernard, 2023; Melzi et al., 2019b; Halimi et al., 2019; Roufosse et al., 2019; Eisenberger et al., 2020b), we evaluate shape matching performance by a commonly used metric, average geodesic errors. Geodesic errors measure the accuracy by calculating the geodesic distance of the matching result w.r.t. the ground truth. For the inter-class benchmark (Abdelreheem et al., 2023a), the average geodesic error is calculated on annotated key-points since only sparse correspondences are labeled.

Table 1: **Experimental results on inter-class shape matching.** The table reports the average geodesic errors of each method on the benchmark SNIS.

| Method | Venue | Year | SNIS |
|---|---|---|---|
| *Functional Map Based* | | | |
| ULRSSM | ToG | 2023 | 0.49 |
| *Rendering Based* | | | |
| Diff3F | CVPR | 2024 | 0.57 |
| ZSC (SEG) | SIGGRAPH Asia | 2023 | 0.41 |
| ZSC (SATR) | SIGGRAPH Asia | 2023 | 0.37 |
| ZSC (SAM-3D) | SIGGRAPH Asia | 2023 | 0.36 |
| DenseMatcher | ICLR | 2025 | 0.28 |
| SeMa3D (ours) | – | – | **0.21** |

## 4.1 PERFORMANCE ON INTER-CLASS SHAPE MATCHING

**Experimental setup.** To test the capabilities for inter-class shape matching, we evaluate our method and baselines on the Strongly Non-Isometric Shapes (SNIS) Dataset (Abdelreheem et al., 2023a). SNIS contains 211 mixed shapes from different sources, including FAUST (Bogo et al., 2014) (human), SMAL (Zuffi et al., 2017) (animals) and DeformingThings4D (Li et al., 2021) (humanoid). For each shape pair, the correspondences are sparsely annotated by 34 keypoints. We compare our method with the representative functional map baseline, ULRSSM (Cao et al., 2023), and semantic feature-based baselines, Diff3F (Dutt et al., 2024), ZSC (Abdelreheem et al., 2023a), and Dense-Matcher (Zhu et al., 2025). Details of the experimental setting can be found in Appendix C.

**Results and analysis.** Table 1 illustrates the experimental results of inter-class shape matching on SNIS benchmark. SeMa3D significantly outperforms both the representative functional map baseline, ULRSSM (0.49), and other semantic feature-based methods. Notably, it surpasses rendering-based approaches like Diff3F (0.57), various versions of ZSC (0.36 − 0.41), and the strong baseline DenseMatcher (0.28). This result underscores the effectiveness of SeMa3D's approach, which integrates visual and linguistic features within a functional map framework enhanced by a region-aware contrastive loss, proving crucial for robustly handling the significant geometric and semantic variations inherent in matching shapes across different categories.

Table 2: **Experimental results on non-isometric benchmarks.** The table reports the average geodesic errors ($\times 100$) of each method on SMAL and TOPKIDS.

| Method | Venue | Year | SMAL | TOPKIDS |
|---|---|---|---|---|
| *Axiomatic Methods* | | | | |
| Smooth Shells | CVPR | 2020 | 36.1 | 11.8 |
| ZoomOut | ToG | 2019 | 38.4 | 33.7 |
| *Functional Map Methods* | | | | |
| UnsupFMNet | CVPR | 2019 | - | 38.5 |
| SURFMNet | ICCV | 2019 | - | 48.6 |
| AttentiveFMaps | NeurIPS | 2022 | 5.4 | 23.4 |
| ULRSSM | ToG | 2023 | 6.0 | 8.9 |
| *Rendering-based Methods* | | | | |
| Diff3F | CVPR | 2024 | 28.4 | 31.0 |
| DenseMatcher | ICLR | 2025 | 4.7 | 6.2 |
| SeMa3D (ours) | – | – | **4.5** | **5.6** |

## 4.2 PERFORMANCE ON NON-ISOMETRIC BENCHMARKS

**Experimental setup.** We evaluate the performance of non-isometric shape matching on SMAL (Zuffi et al., 2017) and TOPKIDS (Lähner et al., 2016). The SMAL dataset contains 49 animals of 8 species, including dog, horse and etc. Following Cao et al. (2023), we use five species for training and the remaining for testing. In this way, only unseen animal categories appear during the inference stage. Another non-isometric dataset, TOPKIDS, focuses on shape matching with topological noise, which is a common phenomenon in real-world 3D data. The presence of topological

noise distorts the intrinsic shape geometry non-isometrically, which severely affects the accuracy of correspondence estimation. The TOPKIDS dataset contains synthetic shapes of children with topological merging based on the outer hull of intersecting shape parts. We use the following baselines: axiomatic methods, Smooth Shells (Eisenberger et al., 2020a) and ZoomOut (Melzi et al., 2019b), functional map-based methods, UnsupFMNet (Halimi et al., 2019), SURFMNet (Roufosse et al., 2019), AttentiveFMaps (Li et al., 2022a), and ULRSSM (Cao et al., 2023), and rendering-based methods, Diff3F (Dutt et al., 2024) and DenseMatcher (Zhu et al., 2025).

**Results and analysis.** Quantitative results are shown in Table 2. Our proposed method consistently outperforms the strong baseline, DenseMatcher, achieving lower average geodesic errors on both SMAL (4.5 vs. 4.7) and TOPKIDS (5.6 vs. 6.2), with improvements that persist under category generalization (testing on unseen animal species) and in the presence of topological noise. SeMa3D's view-consistent colorization stabilizes multi-view descriptors, the fusion of SD-DINO visual features with SigLip language embeddings injects high-level part semantics, and the region-aware contrastive loss enforces part-to-part alignment while tolerating zero-shot segmentation noise. Combined with coupling between soft pointwise maps and functional maps, these design choices yield smoother and more globally consistent correspondences, explaining the observed edge over DenseMatcher and the larger margins over Diff3F and functional-map baselines.

Table 3: **Experimental results on near-isometric benchmarks.** Average geodesic errors ($\times 100$) on FAUST, SCAPE, and SHREC19.

| Method | Venue | Year | **FAUST** | **SCAPE** | **SHREC19** |
|---|---|---|---|---|---|
| *Axiomatic Methods* | | | | | |
| BCICP | ToG | 2018 | 6.4 | 11.0 | 8.0 |
| ZoomOut | ToG | 2019 | 6.1 | 7.5 | 7.8 |
| Smooth Shells | CVPR | 2020 | 2.5 | 4.7 | 7.6 |
| *Functional Map Methods* | | | | | |
| UnsupFMNet | CVPR | 2019 | 4.8 | 9.6 | 11.1 |
| SURFMNet | ICCV | 2019 | 2.5 | 6.0 | 4.8 |
| ULRSSM | ToG | 2023 | 1.6 | 1.9 | 5.7 |
| *Rendering-based Methods* | | | | | |
| Diff3F | CVPR | 2024 | 20.7 | 22.1 | 26.3 |
| DenseMatcher | ICLR | 2025 | 1.6 | 2.0 | 3.1 |
| SeMa3D (ours) | – | – | **1.6** | **1.9** | **3.1** |

### 4.3 PERFORMANCE ON NEAR-ISOMETRIC BENCHMARKS

**Experimental setup.** We evaluate our method on three common near-isometric benchmarks: FAUST (Bogo et al., 2014), SCAPE (Anguelov et al., 2005), and SHREC19 (Melzi et al., 2019a). Following the previous works (Cao & Bernard, 2023), we adopt the remeshed version of them (Ren et al., 2018; Donati et al., 2020). The FAUST dataset contains 100 subjects with 10 persons in 10 poses. We use 80 shapes for training and 20 shapes for evaluation. The SCAPE dataset contains 71 poses of one subject, which is split into 51 shapes for training and 20 shapes for evaluation. The SHREC19 dataset collects 44 human shapes with different poses and styles. We use the following baselines: axiomatic methods, BCICP (Ren et al., 2018), ZoomOut (Melzi et al., 2019b), Smooth Shells (Eisenberger et al., 2020a), functional map-based methods, UnsupFMNet (Halimi et al., 2019), SURFMNet (Roufosse et al., 2019) and ULRSSM (Cao et al., 2023), and rendering-based methods, Diff3F (Dutt et al., 2024) and DenseMatcher (Zhu et al., 2025).

**Results and analysis.** Table 3 summarizes the quantitative results of near-isometric shape matching. For near-isometric shape matching, SeMa3D achieves the best performance, achieving 1.6 on FAUST (tying ULRSSM and DenseMatcher), 1.9 on SCAPE (tying ULRSSM and slightly improving over DenseMatcher at 2.0), and 3.1 on SHREC19 (matching DenseMatcher and substantially outperforming ULRSSM at 5.7 and other baselines). These results illustrate that, despite the performance gain compared with the second-best, DenseMatcher, is relatively slight, introducing semantic features, and the region-aware contrastive loss does not compromise classical near-isometric accuracy; instead, the functional-map coupling and spectral regularization preserve smooth, globally consistent mappings while the view-consistent semantic descriptors remain compatible with isometry-based constraints, yielding strong performance across standard human benchmarks.

Table 4: **Ablation study on novel components.** Average geodesic errors ($\times 100$) comparing different design choices on SMAL and TOPKIDS.

| Method | SMAL | TOPKIDS |
|---|---|---|
| SeMa3D (full) | **4.5** | **5.6** |
| *Colorization* | | |
| FMNet + SD-DINO (w. colorization) | 4.9 | 5.9 |
| FMNet + SD-DINO (w/o colorization) | 7.8 | 10.5 |
| *Semantic Features* | | |
| FMNet + WKS | 27.3 | 11.2 |
| FMNet + XYZ | 6.0 | 8.9 |
| FMNet + SD-DINO | 4.9 | 5.9 |
| FMNet + SD-DINO + SigLip | 4.6 | 5.7 |
| *Loss Function* | | |
| SeMa3D w/o contrastive loss | 4.6 | 5.7 |

## 4.4 ABLATION STUDY

**Experimental setup.** To evaluate the effectiveness of our novel components on the non-isometric setting, we conducted ablation studies on SMAL and TOPKIDS. We use different settings to test the effectiveness of three aspects, respectively: 1) **Colorization:** To test the effectiveness of texture synthesis, this branch contains two variants: the functional map framework with the loss function Equation 6 based on visual semantic features from SD-DINO, with colorization and without colorization, respectively. 2) **Semantic features:** To test the effectiveness of semantic features, this branch contains four variants based on the functional map framework with the loss function Eq. 6, using different features: WKS, XYZ (vertex coordinates), SD-DINO, and SD-DINO plus SigLip. 3) **Loss function:** We use the variant that disable the contrastive loss of SeMa3D.

**Results and analysis.** The experiment results of the ablation study are listed in Table 4. The results highlight three novel components of SeMa3D's gains from different perspectives: (i) the performance gain of colorization confirms that high-quality view-consistent textures are essential for stable multi-view lifting; (ii) semantic feature design matters, with purely geometric cues (like WKS and XYZ) underperforming (27.3/11.2 and 6.0/8.9), SD-DINO providing strong visual cues (4.9/5.9), and adding SigLip language embeddings yielding consistent improvements over the visual semantic features (4.6/5.7), validating the benefit of high-level part semantics; and (iii) the region-aware contrastive loss offers a modest but reliable gain, indicating that explicitly enforcing part-to-part alignment complements functional-map regularization.

## 5 CONCLUSION

In this paper, we introduce SeMa3D, a novel framework to address the challenging settings of non-isometric and inter-class shape matching where conventional functional map methods fail. In summary, we make the following contributions: (i) leverage vision-language features to overcome non-isometric matching limitations of conventional functional maps, (ii) use a colorization strategy for view-consistent semantic descriptors, (iii) augment visual features with text embeddings to inject higher-level semantics, and (iv) employ a region-aware contrastive loss to enforce part-to-part alignment for robust inter-class and non-isometric correspondence.

## ETHICS STATEMENT

- **Dataset and privacy.** In this paper, we only use publicly available datasets (FAUST, SCAPE, SHREC19, SMAL, TOPKIDS, DT4D, SNIS) under their respective licenses. No personal identifiable information is present; human and animal meshes are synthetic or de-identified. We do not collect new human or any animal data, and no human subjects research was conducted.

- **Data collection and labeling.** Our method is purely unsupervised and relies on no annotations; no paid annotators were needed or employed. When using third-party tools, we adhere to their licenses and attribution requirements.

- **Reproducibility.** We will release full code, configuration files, and checkpoints to facilitate reproduction after acceptance. Implementation details are described in Appendix B.
- **Conflicts of interest.** The authors declare no conflicts of interest related to this work.

## REPRODUCIBILITY STATEMENT

- **Code release.** We will release the full codebase after acceptance, including training and evaluation scripts, configuration files, and pre-trained checkpoints. All experiments can be reproduced from a single entry script with the provided configs, seeds, and checkpoints.
- **Dataset fetching.** We only use publicly available datasets. They are publicly downloadable on the Internet.
- **Randomness control.** We use a fixed seed for all experiments, enable deterministic flags where applicable, and disable non-deterministic kernels.
- **Training details.** Implementation details are described in Appendix B.

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

## A  FUNCTIONAL MAP DETAILS

**Functional map pipeline.**  The functional map framework is a method to represent point-wise correspondences in the eigen-function space (Ovsjanikov et al., 2012). Consider the input shape pair $\mathcal{X}$ and $\mathcal{Y}$ represented as triangle meshes, with $n_x$ and $n_y$ vertices, respectively. Later, the first $k$ eigen-functions of the Laplace-Beltrami operator (Vallet & Lévy, 2008) for each shape are computed to compose the reduced functional space. The optimal functional map $C_{xy}$ is computed by the two following loss functions:

$$\mathcal{L}_{\text{fmap}} = \mathcal{L}_{\text{data}} + \lambda_{\text{reg}}\mathcal{L}_{\text{reg}}. \tag{8}$$

The first component $\mathcal{L}_{\text{data}}$ is the data regularization that enforces preservation of descriptor functions:

$$\mathcal{L}_{\text{data}} =\parallel C\mathbf{A}_x - \mathbf{A}_y \parallel^2, \tag{9}$$

where $\mathbf{A}_x$ and $\mathbf{A}_y$ are low-dimensional projections of the two input descriptors onto the eigen-basis. In previous studies of unsupervised shape-matching (Li et al., 2022a; Cao & Bernard, 2023; Cao et al., 2023), as a common setting, the input descriptors are provided by a dedicated feature extractor. Besides, the second component is a pre-defined regularization. Following SURFMNet (Roufosse et al., 2019), two regularization terms are imposed:

$$\mathcal{L}_{\text{reg}} = \lambda_{\text{bij}}\mathcal{L}_{\text{bij}} + \lambda_{\text{orth}}\mathcal{L}_{\text{orth}}, \tag{10}$$

where $\mathcal{L}_{\text{bij}} =\parallel C_{xy}C_{yx} - I \parallel_F^2 + \parallel C_{yx}C_{xy} - I \parallel_F^2$ and $\mathcal{L}_{\text{orth}} =\parallel C_{xy}^\top C_{yx} - I \parallel_F^2 + \parallel C_{yx}^\top C_{xy} - I \parallel_F^2$ enforce the bijectivity and the orthogonality of $C$ respectively.

The functional map framework is a powerful tool for solving the shape-matching problem, which is first introduced in (Ovsjanikov et al., 2012). Recently, a series of innovations have extended it to fit the unsupervised setting with additional regularization and modules (Litany et al., 2017; Roufosse et al., 2019; Halimi et al., 2019; Donati et al., 2020). In this paper, we adopt SURFMNet (Roufosse et al., 2019) as the implementation of the functional map framework. The basic learning pipeline contains the following four steps:

1. Compute the first $k$ eigen-functions of the Laplace-Beltrami operator (Vallet & Lévy, 2008) for each shape. The eigen-functions are used as the spectral embeddings, which are noted as $\Phi_x \in \mathbb{R}^{n_x \times k}$ and $\Phi_y \in \mathbb{R}^{n_y \times k}$ respectively.

2. Compute descriptor functions (e.g., HKS (Sun et al., 2009), WKS (Aubry et al., 2011), SHOT (Salti et al., 2014), or features by dedicated networks (Wang et al., 2019; Sharp et al., 2022)) on each shape, which is expected to be approximately preserved by the underlying map. They are denoted as $\mathcal{F}_x \in \mathbb{R}^{n_x \times d}$ and $\mathcal{F}_y \in \mathbb{R}^{n_y \times d}$ respectively, where $d$ is the dimension of descriptors. After projecting the descriptors onto the eigen-basis, the results are stored as matrices $\mathbf{A}_x \in \mathbb{R}^{k \times d}$ and $\mathbf{A}_y \in R^{k \times d}$ separately.

3. Compute the optimal functional map $C \in \mathbb{R}^{k \times k}$ by solving:

$$C_{\text{opt}} = \arg\min_C \mathcal{L}_{\text{data}}(C) + \lambda\mathcal{L}_{\text{reg}}(C), \tag{11}$$

   where $\mathcal{L}_{\text{data}}(C) =\parallel C\mathbf{A}_x - \mathbf{A}_y \parallel^2$ is the data regularization that enforces preservation of descriptor functions, and $\mathcal{L}_{\text{reg}}(C)$ is the regularization term that imposes some spectral properties, which contains multiple choices (Ovsjanikov et al., 2012).

4. Compute the point-to-point map by converting the functional map, conventionally using nearest neighbor searching between the aligned spectral embeddings $\Phi_x C^\top$ and $\Phi_y$ as a post-processing step.

**Coupling loss.**  To improve robustness under non-isometric deformations, Cao et al. (2023) augments conventional functional map losses (Cao et al., 2023; Sun et al., 2023; Cao & Bernard, 2023) with a coupling loss. This term enforces consistency between the functional map and its counterpart converted from descriptor functions:

$$\mathcal{L}_{\text{couple}} =\parallel C_{xy} - \Phi_y^\dagger \Pi_{yx}\Phi_x \parallel_F^2, \tag{12}$$

where $\Pi_{yx}$ is the soft correspondence by calculating the similarity of input features, and $\dagger$ indicates the Moore–Penrose inverse. Since $\Pi_{yx}$ can be interpreted as a valid pointwise map, the alignment

between the two terms in Eq. (12) suggests potential cycle consistency (Huang et al., 2014), which is a strong regularization for shape matching. The soft correspondence of two input shapes can be calculated by the cosine similarity of the pair of $\mathcal{F}_{\text{out}}$:

$$\Pi_{xy} = \texttt{Softmax} \left( \frac{\mathcal{F}_{\text{out}}^x \cdot \mathcal{F}_{\text{out}}^y}{\| \mathcal{F}_{\text{out}}^x \| \cdot \| \mathcal{F}_{\text{out}}^y \|} / \tau \right), \tag{13}$$

where $\tau$ is the temperature parameter to control the softness of the correspondence matrix.

## B  IMPLEMENTATION DETAILS

**Multi-view rendering.** The multi-view rendering is implemented using PyTorch3D framework (Ravi et al., 2020). In this paper, we render 9 views of the input shape with uniformly distributed elevation and azimuthal angles between $0°$ to $360°$. Each shape is centered around the origin point and normalized to be inside a unit sphere.

**Semantic segmentation.** We use SATR (Abdelreheem et al., 2023b) to perform zero-shot 3D segmentation. The rendered images, the category prompt along with semantic region names are processed by GLIP (Li et al., 2022b) to generate bounding boxes according to text prompts. In this paper, we use simple category prompts for all experiments, e.g. "human" for FAUST, SCAPE, and SHREC19, and "kid" for TOPKIDS. The semantic region proposal is shared for all experiments. Following the setting from Abdelreheem et al. (2023a), the semantic regions of human includes "head", "arm", "hand", "torso", "leg" and "foot", and the semantic regions of animals includes "head", "leg", "foot", "body" and "tail". For inter-class shape matching, we consider matching semantically consistent parts, i.e., humans' "arm" and "leg" to animals' "leg", humans' "torso" to animals' "body", and humans' "hand" and "foot" to animals' "foot". Unmatched parts, e.g., tails, are considered as negatives for all other parts.

**Functional map architecture.** Similarly to Cao et al. (2023), we use DiffusionNet (Sharp et al., 2022) as a feature adapter. We incorporate the geometric feature, wave kernel signatures (WKS), with our semantic features. The dimension of the WKS is set as 128. We choose the first 200 eigenfunctions of the Laplacian matrices as the spectral embeddings. The dimension of the output features of DiffusionNet is set as 256. We use the AdamW optimizer with a learning rate equal to $10^{-3}$ in all experiments.

**Computation and resources.** All experiments were conducted using the PyTorch framework (Paszke et al., 2019) on a Linux server equipped with a single NVIDIA RTX 4090 GPU.

## C  DETAILS OF EXPERIMENTAL SETUP

For fair comparison, we keep the same architecture, hyperparameters, and settings of the functional map framework for our method, ULRSSM, and DenseMatcher. Besides, *we disable the test-time adaptation technique used in ULRSSM for fairness considerations*. For the implementation of DenseMatcher, we use the semantic regions obtained by our method since the original paper needs part annotations.

## D  SENSITIVITY WITH SEGMENTATION AND TEXTURE QUALITY

To understand how much SeMa3D depends on the quality of the input segmentation and synthesized textures, we conduct an additional experiment that perturbs each component in isolation and reports the resulting correspondence accuracy in Table 5. The goal is to verify that SeMa3D does not rely on high-quality segmentation or textures and is robust to different noises.

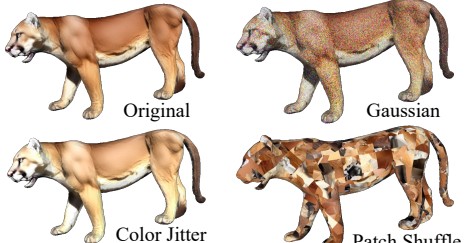

Figure 6: Different perturbations.

Table 5: **Sensitivity with segmentation and texture quality.**

| Method | SNIS |
|---|---|
| GT | 0.199 |
| SeMa3D (proposed) | 0.209 ($\downarrow$ .010) |
| SeMa3D (coarse) | 0.222 ($\downarrow$ .023) |
| *Distorted Segmentation* | |
| Shuffle (5%) | 0.232 ($\downarrow$ .033) |
| Shuffle (10%) | 0.234 ($\downarrow$ .035) |
| Shuffle (20%) | 0.236 ($\downarrow$ .037) |
| *Distorted Textures* | |
| Gaussian | 0.201 ($\downarrow$ .002) |
| Color Jitter | 0.256 ($\downarrow$ .057) |
| Patch Shuffling | 0.245 ($\downarrow$ .046) |

For segmentation, the first row ("GT") uses the coarse ground-truth semantic regions provided by SNIS (Abdelreheem et al., 2023a), where only coarser-level parts such as head, torso, leg, and tail (for animals) or head, torso, leg, and arm (for humans) are available. To this end, we also perform our method on this coarse part proposal that replaces SeMa3D's setting with this coarse part proposal, named "SeMa3D (coarse)". To examine the robustness of SeMa3D under different part proposals, we also include the result of the part proposal used in this paper (same as Section 4.1), named "SeMa3D (proposed)". The "Distorted Segmentation" variants randomly shuffle a given fraction of ground truth segmentation labels to mimic systematic segmentation errors. For texture, the "Distorted Textures" rows keep the segmentation fixed but degrade the synthesized appearance using three independent perturbations—additive Gaussian noise, color jitter, and random patch shuffling—to simulate noisy rendering or imperfect texture synthesis. Different perturbations are illustrated in Figure 6.

Table 5 shows that using SeMa3D's coarse part proposals achieves performance close to that with ground-truth segmentation, indicating that the method is robust to both noisy segmentation and different part proposals. As the proportion of shuffled regions increases, correspondence quality degrades steadily, confirming that the model remains robust to segmentation errors, even severe corruption (20%). Besides, adding Gaussian noise, color jitter, or patch-level shuffling to the textures only causes minor drops in accuracy, suggesting that the proposed feature extraction and multi-view aggregation are still robust under reasonable appearance perturbations. It worthy noting that, all the corrupted variants of SeMa3D consistently outperform uncorrupted baselines (e.g., DenseMatcher of 0.28).

## E  DETAILS OF COMPUTATIONAL EFFICIENCY

Table 6: Computational time and memory consumption of each component.

| | Time | Peak Memory |
|---|---|---|
| Colorization | $\approx 30$ s | 6 GB |
| Segmentation | $\approx 80$ s | 5 GB |
| SD-DINO Feature | $\approx 5$ s | 19 GB |
| Inference (per-pair) | $< 1$ s | 3 GB |

We summarize the computational time and memory consumption of each component in Table 6. Compared with previous semantic correspondence pipelines such as ZSC (Abdelreheem et al., 2023a) and DenseMatcher (Zhu et al., 2025), the overall computational burden of SeMa3D is moderate and compatible with a single high-end GPU. The most time-consuming components are the offline pre-processing stages: texture synthesis ($\approx 30$ s, 6 GB) and zero-shot segmentation ($\approx 80$ s, 5 GB) per shape, both of which are executed once per shape and then reused for all subsequent pairs. In contrast, the online part of our pipeline is lightweight: SD-DINO feature extraction takes about 5 s with a peak memory of 19 GB, and the inference for a single shape pair runs in roughly 1

s using about 3 GB of GPU memory, which fits comfortably on a normal computation device, e.g., a single RTX 4090.

## F  THE USE OF LARGE LANGUAGE MODELS

We employed large language models solely as writing assistants for grammar checking, wording clarification, and light polishing. All technical ideas, method design, experiments, result analysis, and reported results were conceived and executed by the authors. Besides, no datasets, annotations, or quantitative results were generated by LLMs.

