# OpenReview forum: "SeMa3D: Lifting Vision-Language Models for Unsupervised 3D Semantic Correspondence"
_ICLR.cc/2026/Conference — Submitted to ICLR 2026_

### Official Review · Reviewer_CWr9 · 2025-10-31

**Soundness:** 2
**Presentation:** 3
**Contribution:** 2
**Rating:** 4
**Confidence:** 3

**Summary:**

The paper tackles dense 3D semantic correspondence under severe non-isometry and inter-class matching scenarios. The authors propose SeMa3D with the following components:
1. synthesizes multi-view consistent and natural texture from generative models
2. lifts multi-view visual features from SD-DINO and fuses them with language embeddings from SigLIP to compose vertex features
3. optimizes a functional-map correspondence, with a region-aware contrastive loss guided by part proposals from pre-trained model.

On a wide range of inter-class, intra-class and non-isometric benchmarks, SeMa3D produces superior performance comparing to various baseline methods.

**Strengths:**

1. The overall paper is with clear problem focus and motivation: to addresses the 3D correspondence in inter-class and non-isometric cases, simple geometric cues alone will usually fail, thus integrating more semantic feature from a wide range of pre-trained models explicitly should always help.

2. The combination of view-consistent colorization, and the fusion of SD-DINO visual features with SigLIP text embeddings seems straightforward and works well.

3. The authors have conducted extensive experiments to compare the proposed method with a wide range of baselines, also did extensive ablation experiments to validate each proposed components.

**Weaknesses:**

The work relies heavily on directly using a combination of V(L)FMs and texturing model. While the performance gain from these components is naturally expected, the authors didn't fully analyze the potential failure modes from these components, e.g. bad textures, occlusion bias, prompt sensitivity. At least some qualitative results and analysis on inconsistent texture or noisy prompts are helpful.

Similarly, part segmentation outputs can be very noisy, while the loss uses margins to tolerate it, the paper lacks some qualitative robustness analysis vs. segmentation quality.

The adopted multi-stage pipeline may be heavy, current paper didn't give a thorough runtime and resource details, also it'd be good to compare this metric to other baselines.

**Questions:**

Mostly see the weaknesses above, some other questions:

1. What are the end-to-end runtimes per pair, and memory requirements? Any caching or down-stream speedups once features are precomputed? What' the cost for train and test, respectively.

2. How sensitive is the overall performance to the specific part list or to SigLIP’s language prompts across datasets?

---

> ### Author Response · Authors · 2025-11-29
> **Response to Reviewer CWr9**
>
> Thank you for your direct feedback and highlighting analysis opportunities. We now answer each question point-by-point and try to kindly solve your concerns.
>
> > `Q1`: Analysis of failure modes of V(L)FMs and texturing.
>
> `A1`: We acknowledge that simply demonstrating average error improvements is not sufficient to fully understand when SyncMVD, SD-DINO, SigLIP, or SATR fail, even though the ablations already show that each component is critical on SMAL and TOPKIDS. In the revision, we add qualitative side-by-side comparisons showing robustness of our method towards bad textures, or misleading segmentations. **The experimental results qualify that our method is robust to the failure of texture synthesis.** Details of the additional experiment is included in `Appendix D` (marked as orange in rebuttal PDF).
>
> | Method                       | **SNIS** |
> | :--------------------------- | :------: |
> | GT                           |  0\.199  |
> | SeMa3D (proposed)            |  0\.209  |
> | SeMa3D (coarse)              |  0\.222  |
> | ***Distorted Segmentation*** |          |
> | Shuffle (5 %)                |  0\.232  |
> | Shuffle (10 %)               |  0\.234  |
> | Shuffle (20 %)               |  0\.236  |
> | ***Distorted Textures***     |          |
> | Gaussian                     |  0\.201  |
> | Color Jitter                 |  0\.256  |
> | Patch Shuffling              |  0\.245  |
>
> > `Q2`: Robustness vs. segmentation quality.
>
> `A2`: The region-aware contrastive loss, with its dynamic margin based on semantic-region distances, is specifically intended to be robust to noisy zero-shot segmentations by avoiding over-penalizing uncertain boundaries while still enforcing part-consistent features where SATR is confident. Same with the last point, **the experimental results show that our method is robust to wrong segmentation results.** Besides, our method is flexible to incorporate newly released VLMs and segmentation models in the future, e.g., DINOv3 and SAM3.
>
> > `Q3`: Pipeline heaviness, runtime, and resources.
>
> `A3`: The pipeline is indeed multi-stage, but many steps are embarrassingly parallel and amortizable: texture synthesis and zero-shot segmentation are run once per shape, feature extraction can be cached, and the functional-map optimization itself is relatively cheap compared to preprocessing. We add a concise runtime and memory summary in `Appendix F` (marked as cyan in rebuttal PDF) and put the summary table as follows (running time and memory comsumption of each component). Compared with previous semantic correspondence baselines, e.g., ZSC (requires segmentation and MLLM prompting) and DenseMatcher (requires segmentation and SD-DINO feature extraction), **the overall computational burden of our method is moderate and the inference time is very fast**.
>
> |                      |  Time  | Peak Memory |
> | :------------------- | :----: | :---------: |
> | Colorization         | ≈ 30 s |    6 GB     |
> | Segmentation         | ≈ 80 s |    5 GB     |
> | SD-DINO Feature      | ≈ 5 s  |    19 GB    |
> | Inference (per-pair) | < 1 s  |    3 GB     |
>
> > `Q4`: Sensitivity to part lists and language prompts.
>
> `A4`: Following prior work on SATR and ZSC, we empirically use simple, dataset-independent part lists to be stable as long as major semantic parts are covered. We observe that SeMa3D achieves consistent gains across FAUST/SCAPE/SHREC19, SMAL, TOPKIDS, and SNIS under this single configuration, suggesting a reasonable degree of prompt and vocabulary robustness. As illustrated by the table in point 1, **we try a coarser part proposal and the performance remains good, suggesting our method is robust to different part proposals.**

---

### Official Review · Reviewer_Pqo4 · 2025-11-01

**Soundness:** 4
**Presentation:** 3
**Contribution:** 3
**Rating:** 6
**Confidence:** 4

**Summary:**

The paper proposes SeMa3D for unsupervised dense semantic correspondence between 3D shapes, targeting non-isometric deformations and inter-class matching where functional-map pipelines struggle. The method lifts multi-view visual features from VFMs, enforces view-consistent colorization, augments descriptors with language embeddings for part names, and optimizes a functional-map objective coupled with a region-aware contrastive loss that encodes part-to-part relations. Experiments show lower geodesic error than prior art on inter-class SNIS and on non-isometric SMAL and TOPKIDS, while matching state of the art on near-isometric FAUST, SCAPE, and SHREC19.

**Strengths:**

- Originality: The paper lifts both visual and linguistic cues from VLMs to 3D surfaces and injects semantic priors through a region-aware contrastive loss, which is a clear step beyond purely geometric or visual-only descriptors.

- Quality: The method is carefully constructed with view-consistent colorization, multi-view back-projection, semantic region proposals, a functional-map objective, and an explicit semantic contrastive loss. Ablations isolate contributions of colorization, SigLip language embeddings, and the contrastive loss.

- Significance: SeMa3D substantially improves inter-class matching on SNIS and offers consistent gains on non-isometric datasets while retaining top performance on classical near-isometric benchmarks, suggesting practical benefits for cross-category and deformable scenarios.

**Weaknesses:**

- Performance relies on high-quality texture synthesis and zero-shot part segmentation. Failure modes from colorization artifacts or segmentation errors are only partially explored.

- The approach assumes a fixed part proposal set. It is unclear how sensitive results are to vocabulary choices, cross-category ambiguities, or missing parts.

**Questions:**

- How robust is SeMa3D to segmentation noise and to different part vocabularies. Please quantify sensitivity to incorrect or missing regions and report failure cases.

- Do results hold on real scanned meshes with holes and partiality, or in downstream tasks like cross-category part transfer or manipulation. Any small scale study would be helpful.

---

> ### Author Response · Authors · 2025-11-29
> **Response to Reviewer Pqo4**
>
> Thank you for your recognition of our method’s originality and for requesting further robustness evaluation. To address your concerns, we conducted additional experiments and try to answer your questions one-by-one.
>
> > `Q1`: Failure modes from texture synthesis and segmentation.
>
> `A1`: We acknowledge that inconsistent textures and noisy prompts can impact performance; as shown in `Figures 3-5` and discussed in `Section 4.4`, our region-aware contrastive loss and soft coupling mitigate such effects. To further check the robustness of our method towards failures of segmentation and texture synthesis, we conduct additional experiments in `Appendix D` (marked as orange in rebuttal PDF). We also put the result table as follows. **The experimental results suggest that our method is robust to the failure of segmentation and texture synthesis.** Besides, it's quite flexible to incorporate newly released VLMs and segmentation models, e.g., DINOv3 and SAM3, to largely improve the overall performance.
>
> | Method                       | **SNIS** |
> | :--------------------------- | :------: |
> | GT                           |  0\.199  |
> | SeMa3D (proposed)            |  0\.209  |
> | SeMa3D (coarse)              |  0\.222  |
> | ***Distorted Segmentation*** |          |
> | Shuffle (5 %)                |  0\.232  |
> | Shuffle (10 %)               |  0\.234  |
> | Shuffle (20 %)               |  0\.236  |
> | ***Distorted Textures***     |          |
> | Gaussian                     |  0\.201  |
> | Color Jitter                 |  0\.256  |
> | Patch Shuffling              |  0\.245  |
>
> > `Q2`: Sensitivity to segmentation noise and part vocabularies.
>
> `A2`: The dynamic-margin contrastive loss is designed to tolerate segmentation noise by stopping repulsion once features for different regions are “far enough” and by scaling the margin with a semantic distance over the region graph, which empirically helps on noisy zero-shot segmentations. **Following the setting of SATR [1], we empirically use a simple, shared vocabulary across datasets to be stable as long as major semantic parts are covered** (e.g., head/arm/hand/torso/leg/foot for humans and head/body/leg/foot/tail for animals), and the strong generalization on unseen SMAL species and cross-category SNIS suggests that the method is not overly sensitive to this particular choice. **The experimental results illustrated in the table above show that the coarser part proposal brings worse performance** (0.22 v.s. 0.21). We abandon to use a fine-grained part proposal because the segmentation performance is not qualified and it's hard to build semantic relations cross categories. Lastly, we update the explanation of cross-category ambiguities and missing parts in `Appendix B` (marked as lime).
>
> > `Q3`: Robustness to different vocabularies and missing regions (question).
>
> `A3`: In practice, SATR can occasionally miss or merge small parts, and in such cases SeMa3D falls back on visual features plus functional-map regularization while still benefiting from the remaining regions, as illustrated by its robust performance on TOPKIDS with topological noise.​ As illustrated by the table above, **we remove specific part types (e.g., hands and foot) to form a coarser part proposal and the performance remains good, suggesting our method is robust to different part proposals.**
>
> > `Q4`: Real scanned meshes with holes and partiality and downstream tasks (question).
>
> `A4`: Current benchmarks already include non-isometric deformations and topology changes (SMAL, TOPKIDS), but not heavy partiality or scanning artifacts. we agree that evaluating on real scans with holes and partial views is an important next step. We will explicitly state this limitation, and such challenges are out of scope of this paper, since we mainly focus on non-isometric and inter-class shape matching for this paper.
>
>
> [1] Abdelreheem, A., Skorokhodov, I., Ovsjanikov, M., & Wonka, P. (2023). Satr: Zero-shot semantic segmentation of 3d shapes. In Proceedings of the IEEE/CVF International Conference on Computer Vision (pp. 15166-15179).

---

### Official Review · Reviewer_8PXT · 2025-11-01

**Soundness:** 3
**Presentation:** 3
**Contribution:** 3
**Rating:** 6
**Confidence:** 3

**Summary:**

The paper proposes SeMa3D, an unsupervised framework for dense 3D semantic correspondence that tackles non-isometric and inter-class matching settings where classic functional-map pipelines struggle. The method synthesizes view-consistent textures for untextured meshes,  extracts multi-view semantic features with SD-DINO, and augments them with SigLIP text embeddings keyed to zero-shot part proposals, and couples these descriptors with a functional-map objective plus a region-aware contrastive loss to inject part-level priors. Across SNIS (inter-class), SMAL/TOPKIDS (non-isometric), and FAUST/SCAPE/SHREC19 (near-isometric), SeMa3D matches or outperforms recent baselines, with especially notable gains on inter-class SNIS. Ablations indicate that view-consistent colorization, adding language features, and the region-aware contrastive term each contribute measurable improvements.

**Strengths:**

1. Concatenating SD-DINO image features with SigLIP text embeddings tied to zero-shot part labels yields descriptors that disambiguate semantically similar but geometrically dissimilar regions—key for inter-class cases.
2. The dynamic-margin formulation encodes distances over a semantic-region graph and complements the functional-map objective, improving robustness to segmentation noise.

**Weaknesses:**

1. Success depends on a chain of external components—view-consistent texturing, multi-view rendering, zero-shot region proposals, SD-DINO, and SigLIP—so errors can cascade; real-world scans with incomplete geometry/texture or unusual categories might degrade performance. The paper could include an analysis of the error accumulations.
2. While SeMa3D shows improvements, part of the edge over DenseMatcher could stem from using SeMa3D’s own zero-shot region proposals to run that baseline.

**Questions:**

DenseMatcher projects multi-view 2D foundation features to meshes, refines with a 3D network, and solves correspondences via functional maps, evaluated on a colored-mesh dataset (DenseCorr3D) for robotic manipulation. What are the biggest differences between SeMa3D and DenseMatcher? What is the performance of SeMa3D on the DenseCorr3D benchmark?

---

> ### Author Response · Authors · 2025-11-29
> **Response to Reviewer 8PXT**
>
> Thank you for highlighting the core contributions and your insightful critique. We now list your concerns and try to solve them one by one.
>
> > `Q1`: Cascading errors across the component chain.
>
> `A1`: We agree that multi-stage systems can propagate errors, particularly with real-world data. In our ablation studies and qualitative analysis, we demonstrate each components bring benefits for the performance, documented in `Section 3.5.2` and ablation `Table 4`. To further examine the robustness towards segmentation and texture errors, we conduct ablation studies in `Appendix D` (marked as orange in rebuttal PDF). We also put the result table as follows. **As shown in experimental results, our method is robust to different segmentation and texture corruption.**
>
> | Method                       | **SNIS** |
> | :--------------------------- | :------: |
> | GT                           |  0\.199  |
> | SeMa3D (proposed)            |  0\.209  |
> | SeMa3D (coarse)              |  0\.222  |
> | ***Distorted Segmentation*** |          |
> | Shuffle (5 %)                |  0\.232  |
> | Shuffle (10 %)               |  0\.234  |
> | Shuffle (20 %)               |  0\.236  |
> | ***Distorted Textures***     |          |
> | Gaussian                     |  0\.201  |
> | Color Jitter                 |  0\.256  |
> | Patch Shuffling              |  0\.245  |
>
> > `Q2`: Fairness of using SeMa3D’s region proposals for DenseMatcher.
>
> `A2`: DenseMatcher was originally evaluated on colored-mesh benchmarks with manual part annotations, which are unavailable in our unsupervised setting. **We clarify in `Appendix C` that for fairness, providing both methods with the same zero-shot SATR region proposals to avoid giving SeMa3D extra supervision**.
>
> > `Q3`: Differences between SeMa3D and DenseMatcher.
>
> `A3`: Compared with DenseMatcher, **our method utilizes language cues (plus visual semantic features) and our dedicated contrastive loss with a soft margin to enforce semantic consistency.** Besides, DenseMatcher focuses on colored meshes for manipulation tasks and manual segmentation labels, whereas SeMa3D explicitly uses zero-shot part segmentation, and colorization for shape matching benchmarks which contain no segmentation labels and color-less.​ Lastly, experimental results (0.21 error our SeMa3D v.s. 0.28 error of DenseMatcher)
>
> > `Q4`: Performance on DenseCorr3D.
>
> `A4`: We have not yet evaluated SeMa3D on DenseCorr3D, primarily because that benchmark is tailored to robot manipulation scenarios with category-specific colored objects and manual part labels that differ from our unsupervised, untextured mesh setting.​ Besides, our method focuses on human and fourlegged animals, where DenseCorr3D only contains daily objects without semantic-rich dense correspondeses. And the dense annotations for evaluation are not available from the public realse of DenseCorr3D.

---

### Official Review · Reviewer_sFAJ · 2025-11-03

**Soundness:** 3
**Presentation:** 3
**Contribution:** 3
**Rating:** 6
**Confidence:** 4

**Summary:**

The paper proposes SeMa3D, a new framework for unsupervised dense semantic correspondence between 3D shapes, especially targeting non-isometric deformations and inter-class matching (e.g., human-to-horse). The core idea is to leverage vision-language models (VLMs) to extract semantic features from both visual and linguistic domains, and integrate them into a functional map framework to establish robust and semantically meaningful 3D correspondences without any manual annotations.

**Strengths:**

The method introduces Vision-Language Models for 3D Correspondence, where they use SD-DINO (visual) and SigLip (linguistic) models to extract semantic features from multi-view renderings. It combines visual and text embeddings to form rich, semantic-aware vertex descriptors.

The method applies SyncMVD for high-quality, cross-view consistent texture synthesis on raw 3D shapes, which ensures stable and consistent multi-view feature lifting.

The method achieves Zero-Shot Semantic Region Proposal by using SATR for zero-shot 3D part segmentation (e.g., head, leg, torso) without manual annotations, enabling semantic region-aware matching.

**Weaknesses:**

The method uses several pretrained models, which may have a strong dependence on their performance. Are there any correction schemes if the pretrained models are inaccurate?

The method is designed for 3D meshes with consistent topology. It is limited to Mesh-Based Shapes and cannot directly apply to point clouds or unstructured 3D data.

The pipeline involves multiple stages: texture synthesis, multi-view rendering, VLM feature extraction, and functional map optimization. It may be slower than end-to-end or purely geometric methods.

There is no explicit handling of severe topological changes. While robust to topological noise (e.g., TOPKIDS), it may struggle with extreme topological variations not seen during training.

**Questions:**

Please answer and discuss the problems in ``Weakness''.

---

> ### Author Response · Authors · 2025-11-29
> **Response to Reviewer sFAJ**
>
> Thank you for your detailed review and constructive observations. We conducted additional experiments and try to address your questions point-by-point.
>
> > `Q1`: Dependence on multiple pretrained models and error correction.
>
> `A1`: We appreciate your concern regarding reliance on VLMs and segmentation models. To mitigate possible inaccuracies, our **region-aware contrastive loss employs a dynamic margin that improves robustness to segmentation noise**—a mechanism that tolerates occasional mislabeling, documented in `Section 3.5.2` and ablation `Table 4`. To further check the robustness of our method towards segmentation and texture quality, we conduct additional experiments in `Appendix D` (marked as orange in rebuttal PDF). We also put the result table as follows. **The experimental results suggest that our method is robust to different segmentation and texture corruption.** Lastly, our framework can easily incorporate brand-new VLMs and segmentation models, e.g., recently proposed DINOv3 and SAM3, to largely improve the overall performance.
>
> | Method                       | **SNIS** |
> | :--------------------------- | :------: |
> | GT                           |  0\.199  |
> | SeMa3D (proposed)            |  0\.209  |
> | SeMa3D (coarse)              |  0\.222  |
> | ***Distorted Segmentation*** |          |
> | Shuffle (5 %)                |  0\.232  |
> | Shuffle (10 %)               |  0\.234  |
> | Shuffle (20 %)               |  0\.236  |
> | ***Distorted Textures***     |          |
> | Gaussian                     |  0\.201  |
> | Color Jitter                 |  0\.256  |
> | Patch Shuffling              |  0\.245  |
>
> > `Q2`: Application on point clouds and other unstructured data.
>
> `A2`: You are correct that our current pipeline is tailored for mesh-based inputs, like all baselines mentioned in this paper. While adaptation to point clouds or unstructured data is outside the current scope, SeMa3D can be flexibly extended to other 3D representations via different eigenfunction calculation and adaptation of functional-maps. We will clarify this limitation and position such extensions as future work, noting that current benchmarks and **related functional-map baselines are predominantly mesh-based**.
>
> > `Q3`: Multi-stage pipeline and efficiency.
>
> `A3`: The heaviest stages (SyncMVD colorization and SATR part segmentation) are one-time pre-processing for each shape, while feature extraction and functional-map optimization are shared across many pairs and can be cached, so test-time correspondence for new pairs is dominated by the relatively lightweight functional-map stage. To clarify this point, we add a brief discussion of runtime and memory in `Appendix F` (marked as cyan in rebuttal PDF) and put the summary table as follows. Compared with previous semantic correspondence pipelines such as ZSC (requires segmentation and MLLM prompting) and DenseMatcher (requires segmentation and SD-DINO feature extraction), **the overall computational burden of our method is moderate and the inference time is very fast**. We also find that the accuracy improvements, particularly in inter-class and non-isometric scenarios, justify the computational trade-off for many applications.
>
> |                      |  Time  | Peak Memory |
> | :------------------- | :----: | :---------: |
> | Colorization         | ≈ 30 s |    6 GB     |
> | Segmentation         | ≈ 80 s |    5 GB     |
> | SD-DINO Feature      | ≈ 5 s  |    19 GB    |
> | Inference (per-pair) | < 1 s  |    3 GB     |
>
> > `Q4`: Handling severe topological changes.
>
> `A4`: Our experiments already include topological noise via TOPKIDS, where SeMa3D outperforms strong baselines, indicating robustness to moderate topology perturbations. However, we agree that extreme and adversarial topology changes (e.g., large missing parts, different genus) are not covered by current benchmarks and such challenges are out of scope of this paper. We remain this challenge and consider solutions in future work.

---

### Author Response · Authors · 2025-11-29
**General Response**

Dear ACs, SACs, and Reviewers,

We kindly thank the thoughtful and detailed comments from all reviewers, whose expertise and effort greatly helped us strengthen the paper and clarify our contributions.

---

We appreciate all reviewers' recognition that SeMa3D tackles **unsupervised dense 3D semantic correspondence specifically in the challenging regimes of strong non-isometry and inter-class matching**, where conventional pipelines and purely geometric descriptors struggle. To best of our knowledge, SeMa3D is the first shape matching framework that addresses inter-class and non-isometric challenges by **lifting both visual and linguistic cues** from vision(-language) foundation models. We believe our work is beneficial for the research community and can inspire studies borrowing semantics and vision langauge models in depth to solve more challenging scenarios of unsupervised semantic 3D correspondence.

In summary, we appreciate positive feedbacks from reviewers, which highlight:

- **Originality in jointly exploiting VLM-based visual and language cues for rich and semantic-aware descriptors** as a clear step beyond geometric or visual-only approaches, and notes the strong and consistent gains on inter-class and non-isometric benchmarks (`R-sFAJ`, `R-8PXT`,  `R-Pqo4`, `R-CWr9`).
- **Effectiveness of view-consistent colorization, multi-view SD-DINO features to ensures stable and consistent multi-view feature lifting** (`R-sFAJ`, `R-8PXT`, `R-Pqo4`, `R-CWr9`).
- **Carefully designed dynamic-margin contrastive loss that encodes semantic-region relations over a semantic-region graph and complements the functional-map objective**, improving robustness to segmentation noise (`R-8PXT`, `R-Pqo4`).
- **Extensive experiments and ablations that dissect the contribution of colorization, semantic features, and the contrastive loss**, suggesting substantially improves inter-class matching and offering consistent gains for non-/near-isometric deformations (`R-Pqo4`, `R-CWr9`).

---

To address concerns raised by reviewers, we have conducted additional experiments and analysis, including:

- **Dependence on multiple pretrained components and robustness to texture quality, segmentation noise and part proposals.** In response, we added targeted robustness studies that systematically corrupt both segmentation and texture, and a coarser part proposal. These experiments show that **SeMa3D remains strong under substantial segmentation, texturing noise and coarser vocabularies, and that even heavily corrupted variants still outperform uncorrupted baselines such as DenseMatcher**.  (`R-sFAJ`, `R-8PXT`, `R-Pqo4`, `R-CWr9`).
- **Pipeline heaviness, runtime, and resource usage.** We added a concise runtime and memory breakdown for each stage (colorization, zero-shot segmentation, feature extraction, and per-pair inference) in Appendix F. The table clarifies that **all the heavy components are one-time, offline preprocessing per shape, and the online per-pair functional-map inference is lightweight** (about 1 s and 3 GB on a single RTX 4090). (`R-sFAJ`, `R-CWr9`).
- **Scope limitations: meshes only, severe topology changes, real scans/partiality, and downstream tasks.** We explicitly clarified in the revision that the current instantiation of SeMa3D **targets mesh-based inputs, like all functional-map baselines considered**, and that adapting the framework to point clouds or other unstructured representations is feasible but beyond the scope of this work and left to future research (`R-sFAJ`, `R-Pqo4`, `R-8PXT`).
- **Fairness and comparison to DenseMatcher, and missing DenseCorr3D evaluation.** To resolve fairness concerns, we clarified that DenseMatcher in our evaluation uses exactly the same SATR zero-shot region proposals as SeMa3D, thereby **avoiding any extra supervision for our method and ensuring that both pipelines operate under the same segmentation input**. We also detail the methodological differences compared to DenseMatcher (`R-8PXT`, `R-CWr9`).

---

To address the concerns of reviewers, we made the following revisions on the draft:

- Added an additional section to examine the influence of segmentation noise, texture quality and part proposals in `Appendix D` (marked as orange).
- Added an additional section to clarify the runtime and memory comsuption of each component in `Appendix E` (makred as blue).
- Included additional explanations for how we tackle part relations of inter-class pairs and missing parts in `Appendix B` (makred as lime).

---

We sincerely thank AC and reviewers for their valuable comments and great efforts. We respectfully hope our responses have adequately addressed all concerns raised by reviewers. Thank you very much!



Best regards,

Authors of paper 7677

---

### Meta-Review · Area_Chair_e4D8 · 2025-12-28

**Summary:**

This paper presents a framework that integrates semantic knowledge from vision-language foundation models to build robust vertex-level descriptors. All reviewers acknowledged that it depends a complex pipeline. Reviewers also questioned other places like topology changes, unclear or lacking ablation, compuational analysis etc.

Some specific concerns: It is limited to Mesh-Based Shapes and cannot directly apply to point clouds or unstructured 3D data. The method does not explicitly address severe topological changes. Although it demonstrates robustness to moderate topological noise (e.g., as shown on TOPKIDS), its performance may degrade under extreme topological variations that are substantially different from those observed during training, limiting its generalization in more challenging real-world scenarios. The method relies on a pipeline of external components, making it susceptible to error propagation across stages.

**Reviewer Concerns:**

reviewers are not engaged in the rebuttal.

**Reviewer Scores:**

NA

---

### Decision · Program_Chairs · 2026-01-26

Reject